# Integrative study of pulmonary microbiome and clinical diagnosis in pulmonary tuberculosis patients

Hongli Sun,[1] Qiuyue Chen,[2] Dong Zhang,[1,3] Long Hu,[4] Song Li,[4] Minya Lu,[1] Yao Wang,[1] Huiting Su,[1] Yi Gao,[1] Jiayu Guo,[1] Ying Zhao,[1] Juan Du,[1] Cun Liu,[5] Han Xia,[4] Yingchun Xu,[1] Xiaojun Ge,[2] Qiwen Yang[1,3]

**ABSTRACT**  This study investigated the diagnostic potential of mNGS for detecting MTB in pulmonary tuberculosis patients. We analyzed pulmonary microbiome data to assess its impact on mNGS diagnostic accuracy and explored the association between microbiome profiles and clinical diagnosis. Bronchoalveolar lavage fluid samples were collected from 236 patients with pulmonary infections, and the diagnostic performance of mNGS was compared with traditional methods in detecting MTB. Furthermore, the incidence of false negatives and false positives, as well as the characteristics of the lung microbiota in TB patients, was analyzed to improve the diagnostic precision of mNGS. We observed that among all detection methods, mNGS showed the highest sensitivity (73.33%), followed by X-pert (60.00%), culture (53.33%), RT-PCR (53.33%), and sputum smear (23.33%). Notably, mNGS produced 3 false positive results in 236 samples, yielding a specificity of 98.54%. Analysis of the pulmonary microbiome revealed significant differences in both α-diversity and β-diversity between patients with TB and uninfected controls (P<0.05). Shannon index and Chao1 index were identified as significant predictors associated with MTB infection. ROC curve analysis demonstrated an AUC of 0.765, indicating good discriminatory performance. This study suggested that integrating wet-laboratory techniques with bioinformatics analysis can further enhance the diagnostic accuracy of mNGS for TB. Furthermore, microbiome analysis holds significant potential for the diagnosis of MTB infection.

**IMPORTANCE**  This study focuses on the application of next-generation sequencing (NGS) technology in detecting *Mycobacterium tuberculosis* in bronchoalveolar lavage fluid and explores the impact of *M. tuberculosis* infection on the pulmonary microbiome. By optimizing the methods and conducting microbial analyses, the accuracy of metagenomic NGS for detecting *M. tuberculosis* has been improved.

**KEYWORDS**  *Mycobacterium tuberculosis*, metagenomic next-generation sequencing, pulmonary microbiome, bronchoalveolar lavage fluid

Tuberculosis (TB), caused by *Mycobacterium tuberculosis complex* (MTBC), remains one of the major infectious diseases worldwide, severely impacting global public health. According to the World Health Organization (WHO), TB ranks among the top 10 causes of high-mortality infectious diseases, leading to millions of cases and deaths annually (1). The burden of this disease is particularly profound in low- and middle-income countries, and its control and prevention have become a focal point of international concern.

Current diagnostic methods for TB, including sputum smear microscopy, culture, and molecular biology methods such as X-pert, have their limitations (2, 3). Sputum smear microscopy is simple to operate but has limited sensitivity and specificity (4); although culture is the gold standard, its long duration limits timely treatment for patients; X-pert is fast and sensitive but costly and still faces challenges in detecting drug-resistant TB.

**Peer Reviewer** Tingting Yang, Zhejiang University School of Medicine, Hangzhou, China

Address correspondence to Qiwen Yang, yangqiwen81@vip.163.com, Xiaojun Ge, gxj_199421@163.com, or Yingchun Xu, xycpumch@139.com.

Hongli Sun, Qiuyue Chen, Dong Zhang, and Long Hu contributed equally to this article. Author order was determined on the basis of seniority.

The authors declare no conflict of interest.

Furthermore, for non-pulmonary TB cases, the efficacy of these traditional methods is further diminished (5).

Metagenomic next-generation sequencing (mNGS), a product of advances in bioinformatics over the past decades, has become a powerful tool for the identification and characterization of microbial pathogens. As one of these tools, mNGS can not only detect known pathogens but also discover new or uncommon ones (6, 7), promising to be an effective diagnostic instrument. Particularly where traditional methods fail to provide a definitive diagnosis or when pathogens are difficult to culture, mNGS has shown its unique advantages.

Beyond pathogen detection, research on the pulmonary microbiome is a burgeoning field in recent years (8–11). The pulmonary microbiome refers to the genetic material of all microorganisms in the lung environment, including bacteria, viruses, fungi, and other microbes (12). The microbiome of a healthy individual's lungs is relatively stable, but it may undergo changes in respiratory diseases such as TB (9, 13). These changes can not only affect disease progression but also serve as biomarkers for disease status (12). Studying alterations in the pulmonary microbiome offers a new perspective in understanding the infection process of MTBC and its impact on the host's pulmonary environment.

The aim of this study was to evaluate the diagnostic performance of the mNGS method for detecting MTBC and to investigate strategies for reducing false-positive and false-negative results. To further enhance the sensitivity of the detection, we conducted a microbiome analysis to compare the microbial community composition between healthy individuals and TB patients, exploring whether these microbial features could serve as auxiliary biomarkers to improve the accuracy of MTBC detection.

By combining traditional pathogen detection methods with mNGS microbiome analysis, we hope to bring new insights into the diagnosis and treatment of TB, while providing scientific evidence and new research directions for global TB control strategies.

## MATERIALS AND METHODS

This study included a cohort of 260 patients who presented with suspected pulmonary infections at Peking Union Medical College Hospital between 1 April 2022 and 30 November 2023. After excluding 11 cases with unclear clinical diagnoses and 13 cases with missing test results, a total of 236 patients were included in a methodological comparison. Based on the diagnostic criteria for TB outlined in WS288-2017 (14), patients were categorized into three groups: the TB group (TB) with 22 cases, the previous TB group (PN) with 12 cases, and a non-specific pulmonary infection group (NC) with 22 cases.

### Sputum

Sputum samples were collected in the morning from each patient over three consecutive days into sterile sputum cups, ensuring a minimum volume of 3 mL. The Ziehl-Neelsen staining method (BASO, Zhuhai) was used under a microscope at 1,000× magnification (100× objective lens and 10× eyepiece), observing 300 fields or the entire sample area to perform acid-fast staining.

### Bronchoalveolar lavage fluid

For each patient, a bronchoscope was inserted into a selected segmental bronchus to perform 2–3 consecutive lavages with 37°C sterile saline solution each using 40–60 mL. The lavage fluid was then aspirated for recovery. Each BALF sample underwent MTBC culture, mycobacterial RT-PCR testing, X-pert MTB/RIF, and mNGS assays.

#### Culture

The BD MGIT960 fully automated mycobacterial culture monitoring system (BD, USA), BBL MGIT culture tubes and reagent kits were used, along with Roche culture medium

(Lowenstein-Jensen). Positive cultures were further confirmed as MTBC following Ziehl-Neelsen staining and MPT64/MPB64 antigen testing.

### PCR testing

Following sample liquefaction, specimens were oscillated using the Bioer GenePure nucleic acid extraction instrument and treated in a 95°C metal bath for 5 min for nucleic acid extraction. A mycobacterial nucleic acid testing kit (Bioer GenePure, Chengdu) was used for the detection of both TB and non-tuberculosis mycobacteria (NTM). The total volume for the assay was 20 µL, performed on a real-time fluorescent quantitative PCR instrument (Bioer GenePure).

### X-pert MTB/RIF assay

One milliliter of BALF was mixed with an equal volume of sample reagent and left at room temperature for 10–15 min to ensure adequate liquefaction. Then, using a new pipette tip, 2 mL of the processed sample was transferred into the X-pert MTB/RIF cartridge and placed into the Gene X-pert Dx system module for automatic testing. Results for MTBC detection were available within 2 h. The X-pert MTB/RIF cartridges and the detection module were provided by Cepheid (USA).

### mNGS

The mNGS detection process includes host cell removal, nucleic acid extraction, library preparation, sequencing, and bioinformatics analysis. Briefly, DNA was extracted using a DNA extraction kit (Tiangen Biotech [Beijing] Co., Ltd., China) in a 1 mL sample. Libraries were constructed for the DNA samples using a Nextera XT DNA Library Prep Kit (Illumina, San Diego, CA). Index PCR parameters were as follows: 1 cycle at 72°C for 3 min and 98°C for 30 s, followed by 17 cycles at 98°C for 15 s, 60°C for 30 s, and 72°C for 30 s, and a final cycle at 72°C for 5 min and 4°C for 10 s. Dual indexing was conducted by employing the IDT for Illumina DNA/RNA UD indexes. The size distribution was measured on the Qsep 1, and the concentration of the libraries was quantified by the Qubit dsDNA HS Assay kit on a Qubit 3.0 fluorometer (Thermo Fisher Scientific, Waltham, MA, USA). Library pools were then loaded onto the Illumina Nextseq CN500 sequencer for 75 cycles of single-end sequencing, generating approximately 20–40 million reads for each library. Sequencing data were processed using Trimmomatic to remove low-quality sequences, sequences shorter than 40 bp, and adaptor sequences to obtain high-quality data. Sequence analysis was performed using Vision Medicals' IDseq commercial bioinformatics pipeline. Reads aligning to the human genome and plasmids were excluded, and the remaining sequences were taxonomically classified by comparison to Vision Medicals' curated microbial database. The interpretation of mNGS reports was conducted by a team from the Pathogen Sequencing Laboratory of Peking Union Medical College Hospital, consisting of professionals in bioinformatics, clinical laboratory diagnostics, and infectious diseases.

### Microbial community structure analysis

We calculated the α-diversity indices of the microbial community structure using the vegan package version 2.5-7 (15), which includes Chao1, Shannon, ACE, and Simpson indices. The α-diversity differences between groups were evaluated using the *t*-test to assess significant variances. A β-diversity distance matrix was computed based on the Bray-Curtis distance algorithm, and Principal Coordinates Analysis (PCoA) was conducted with the ape package in R software version 4.1.0 (16). Differences in community structures were further assessed for statistical significance using (15) permutational MANOVA.

The LEfSe (17) (Linear discriminant analysis Effect Size) method was employed to identify biomarkers with significant differences between biological conditions. First, features with significant abundance differences were determined using the

non-parametric Kruskal-Wallis (KW) rank-sum test, and species with significant discrepancies were identified. Subsequently, the impact of species abundance differences on intergroup differences was evaluated using Linear Discriminant Analysis (LDA), and species with an LDA score greater than 2 or 4 were selected as characteristic dominant species for each group. Wilcoxon rank-sum tests and KW tests were used to analyze significant differences in species between two groups and multiple groups, respectively, with a significance threshold of $P < 0.05$.

Based on the method by Mac et al. (11), we established a microbial interaction network by first filtering genera with an abundance of at least 0.01 in a minimum of 5% of the samples and recalculating their relative abundances. We assessed the similarities between microbes using the Reboot method described by Faust and others (18), utilizing mutual information, Pearson and Spearman correlation coefficients, Bray-Curtis similarity, and Generalized Enhanced Linear Models (GBLMs), along with bootstrap methods and renormalization strategies to identify correlations between microbes. The Mann-Whitney *U*-test was used to compare the null distribution with the bootstrap distribution, and the *P*-values and scores for network edges were integrated using the Simes test. Finally, the co-occurrence network was generated and visualized using Cytoscape software version 3.7.2 (19) and the Diffany (20) plugin, retaining only network edges with an FDR value less than 0.001.

## Statistical analysis

Numpy and scipy.stats in Python v3.11.4 was used for statistical analyses. Data for continuous variables were expressed as the mean ± standard deviation (SD). The Shapiro–Wilk test is used to determine whether data follows a normal distribution, *P*-value > 0.05 indicates that the data may follow a normal distribution. Data for categorical variables (%) were compared using the chi-square test, and *P*-value < 0.05 was considered statistically significant. Use the UpSetR package in R v4.3.1 to create an UpSet plot, ggplot2 to create a bar chart, pROC to create ROC. To identify the most relevant predictors for the detection outcome, we performed Least Absolute Shrinkage and Selection Operator (LASSO) logistic regression using the glmnet package in R. The optimal lambda parameter was determined using 10-fold cross-validation. The predictive performance of the selected variable was assessed using receiver operating characteristic (ROC) curve analysis. The area under the curve (AUC) was calculated to evaluate its discriminatory power, with values closer to 1.0 indicating better performance.

## RESULTS

In this study, a total of 236 BALF samples were analyzed, as depicted in Fig. 1. The age distribution of the participants ranged from 13 to 94 years, with an average age of 55 years (median and IQR: 55 years, IQR 44–66 years), of which 133 were male (56.36%). Detailed demographic information of the patients is presented in Table 1. After thorough clinical assessment, a total of 30 patients (representing 12.71% of the sample size) were diagnosed with TB. Within these 30 confirmed cases, 23 individuals were verified as infection-positive through etiological testing. The remaining seven patients, despite negative etiological test results, showed significant symptom improvement following standardized anti-TB treatment and exhibited treatment responses in subsequent radiological examinations, thus confirming the clinical diagnosis.

## Diagnostic performance comparison

For the detection rates of MTBC, no significant statistical differences were observed among mNGS, culture, PCR, and X-pert (*P > 0.05*); however, all were significantly superior to sputum smear tests (detection rates were 10.59% compared to 2.97%, *P < 0.05*), as detailed in Fig. 2. Based on the clinical composite diagnosis, 30 cases diagnosed with pulmonary TB are summarized in Fig. S1, showing the results of different testing methods. The sensitivity of mNGS for detecting pulmonary TB was significantly higher

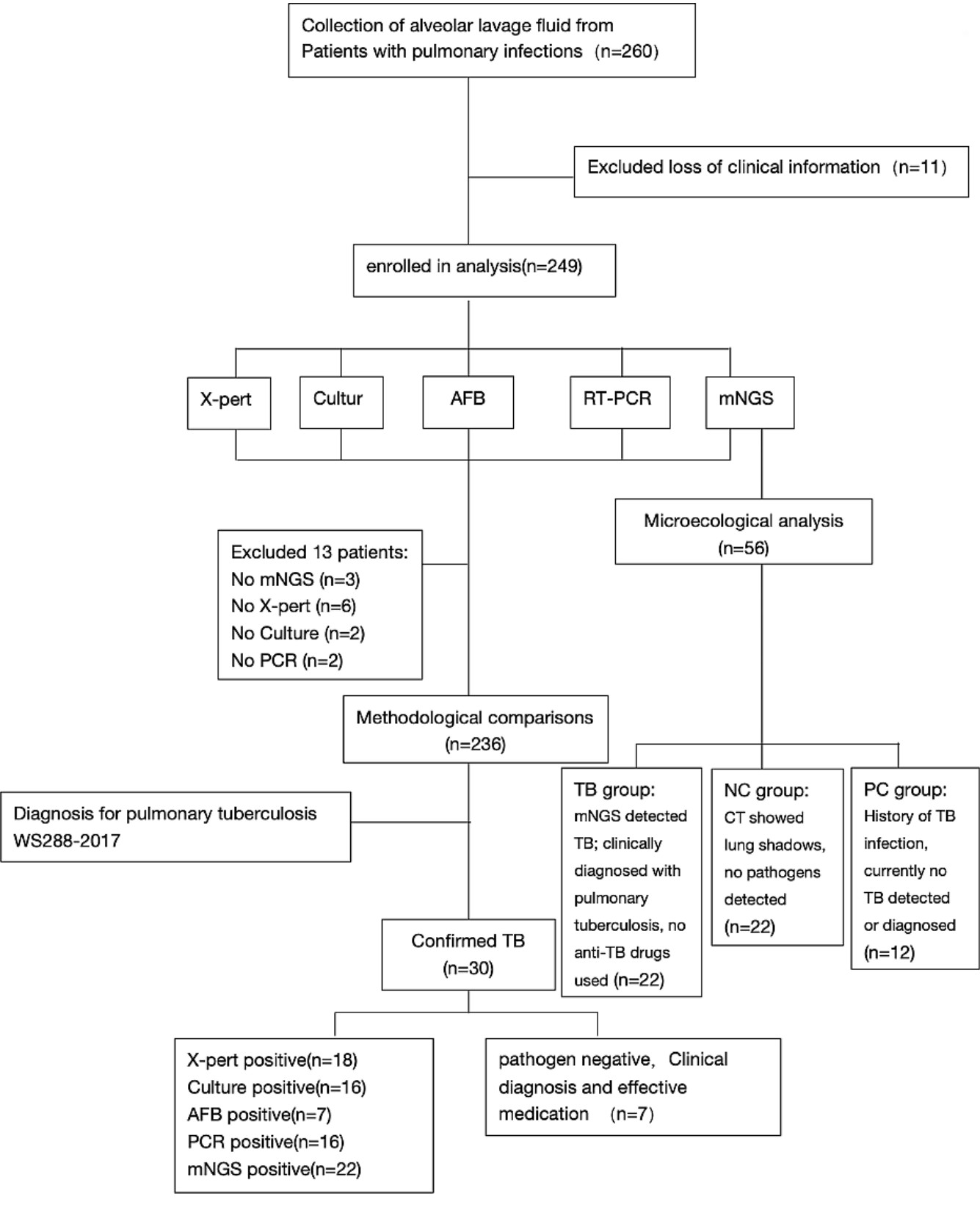

**FIG 1** Workflow of this study.

**TABLE 1** Baseline characteristics of the study population[a]

| Characteristic | Clinical value | P-value |
|---|---|---|
| Age (years) | | |
| Median (IQR) | 55 (44, 66) | P > 0.05 |
| Sex | | |
| Male (%) | 133 (56.4) | |
| Female (%) | 103 (43.6) | |
| Others | | |
| WBC ($10^9$/L) | 7.9 ± 4.3 | P < 0.05 |
| CRP | 56.6 ± 66.1 | P > 0.05 |
| Clinical characteristic (individuals) | | |
| History of TB | 13 | |
| NTM | 11 | |
| Deaths | 5 | |

[a]The Shapiro-Wilk test was used to assess if the continuous variables conform to a normal distribution, $P < 0.05$ was considered to indicate a non-normal continuous variable. WBC, white blood cell count; CRP, C-reactive protein. NTM, non-tuberculous *Mycobacterium*.

than that of the sputum smear (73.33% compared to 23.33%, *P < 0.05*), as shown in Table 2. The specific results detected by each method are shown in Fig. S2.

In the mNGS detection, eight false-negative cases were observed. Among these, three cases involved individuals who had undergone anti-TB treatment. The remaining five individuals have unclear circumstances. Additionally, three false-positive cases were detected. Furthermore, two samples initially tested negative but yielded positive results upon secondary testing following the application of specialized experimental procedures. The detailed classification of these findings is presented in Fig. 3.

## Pulmonary microbiome analysis

Utilizing mNGS technology, we revealed the differences in the pulmonary bacterial community between different groups. We analyzed the samples from 22 cases with no significant lung infection (NC group), 22 patients positive for TB (TB group), and 12 cases with clinically outdated TB (PN group). Significant differences in α-diversity were observed among the three groups (*P < 0.05*) in Fig. 4. β-diversity analysis through PCoA indicated a substantial separation of the pulmonary microbiota between the TB group and both the NC and PN groups, as shown in Fig. 5 and Fig. S3. Within the three groups, there are certain differences in the average abundance of the top 10 species and genera, as shown in Fig. 4. For instance, there are significant differences at the species level, such as with MTBC and Propionibacterium acnes, while at the genus level, differences include MTBC, *Rothia, Veillonella,* and *Burkholderia.*

We identified factors that showed significant differences between the TB group and NC group, with a focus on the following microbial species: *Streptococcus parasanguinis, Rothia mucilaginosa, Streptococcus mitis, Prevotella melaninogenica,* as well as the Shannon index and Chao1 index. Through Lasso regression analysis, we selected two factors that were significantly associated with the clinical diagnosis of TB: the Shannon index and Chao1 index. Based on these significant factors, we constructed the following assessment index:

$$Y = -0.004 \times \text{Shannon index} + 0.313 \times \text{Chao1 index}$$

We then used this assessment index to plot the ROC curve. The area under the ROC curve (AUC) was 0.765, indicating that the model has a moderate ability to discriminate between groups. Based on the ROC curve, we determined the cut-off value to be 112.96, which serves as the threshold for clinical diagnosis of TB, as shown in Fig. 6. Using this cut-off value, we analyzed the above five false-negative cases of unknown cause and identified two cases of MTBC infection; the details are shown in Table 3.

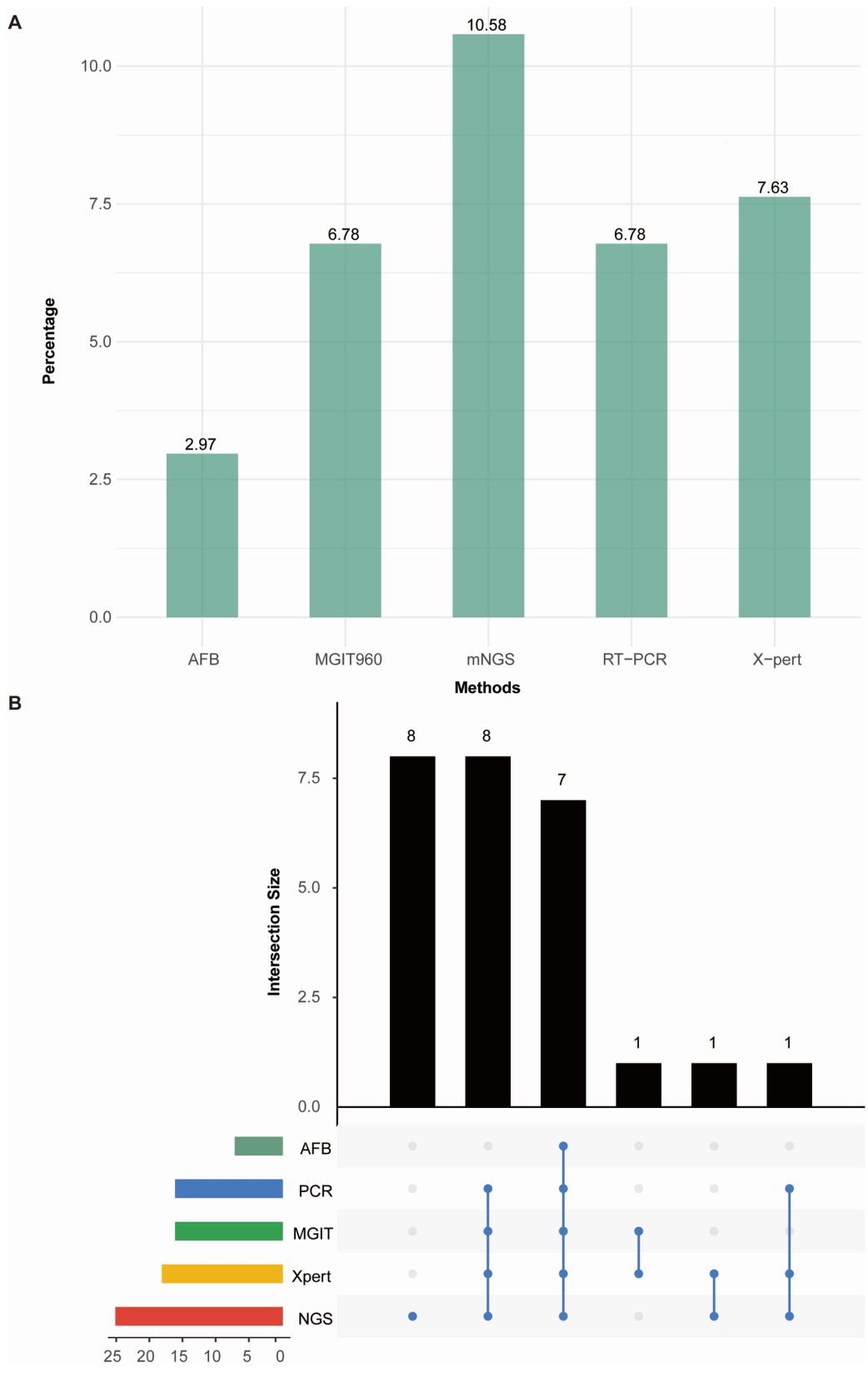

**FIG 2**  (A) Detection rates of *M. tuberculosis* by AFB, MGTI960, mNGS, RT-PCR, and Xpert MTB/RIF. (B) Detection results for five different methods.

**TABLE 2** Diagnostic efficacy of AFB, MGIT960, PCR, Xpert MTB/RIF, and mNGS[a]

| Method | Sensitivity (95% CI) | Specificity (95% CI) | PPV (95% CI) | NPV (95% CI) |
|---|---|---|---|---|
| AFB | 23.33% (8%–38%) | 100% (100%–100%) | 100% (100%–100%) | 90% (86%–94%) |
| MGIT960 | 53.33% (35%–71%) | 100% (100%–100%) | 100% (100%–100%) | 94% (90%–97%) |
| PCR | 53.33% (35%–71%) | 100% (100%–100%) | 100% (100%–100%) | 94% (90%–97%) |
| Xpert MTB/RIF | 60.00% (42%–78%) | 100% (100%–100%) | 100% (100%–100%) | 94% (91%–98%) |
| mNGS | 73.33% (58%–89%) | 98.54% (97%–100%) | 88% (75%–101%) | 96% (94%–99%) |
| Five methods[b] | 76.67% (62%–92%) | 98.54% (97%–100%) | 88% (76%–101%) | 97% (94%–99%) |

[a]CI, confidence interval; PPV, positive predictive value; NPV, negative predictive value.
[b]The five methods are AFB, MGIT960, PCR, Xpert MTB/RIF, and mNGS.

We further conducted co-occurrence analysis, and the results prominently demonstrated significant variations in microbial interactions under different states. Compared to the NC and PN groups, the TB group exhibited increased numbers and intensities of positive microbial interactions. To thoroughly evaluate microbial interactions associated with MTBC infection, we performed network differential analysis, using the NC group as the reference and TB and PN groups as the conditional groups, to identify consensus networks among the three groups. This analysis revealed conservative microbial communities commonly present across all three groups, as detailed in Fig. 7.

## DISCUSSION

In recent years, mNGS technology has been widely applied in the detection of pathogenic microorganisms, with numerous studies reporting its application in TB detection (20–22). Our study found that although mNGS has a higher overall detection rate for MTBC compared to other methods, it was not statistically significant but was significantly superior to sputum smear microscopy ($P < 0.05$). Moreover, in our study, mNGS identified 11 cases of NTM, including five cases of *Mycobacterium avium complex*, four cases of *Mycobacterium abscessus*, and two cases of *Mycobacterium intracellulare,* suggesting that it is also an excellent method for detecting NTM, whose prevalence has been increasing

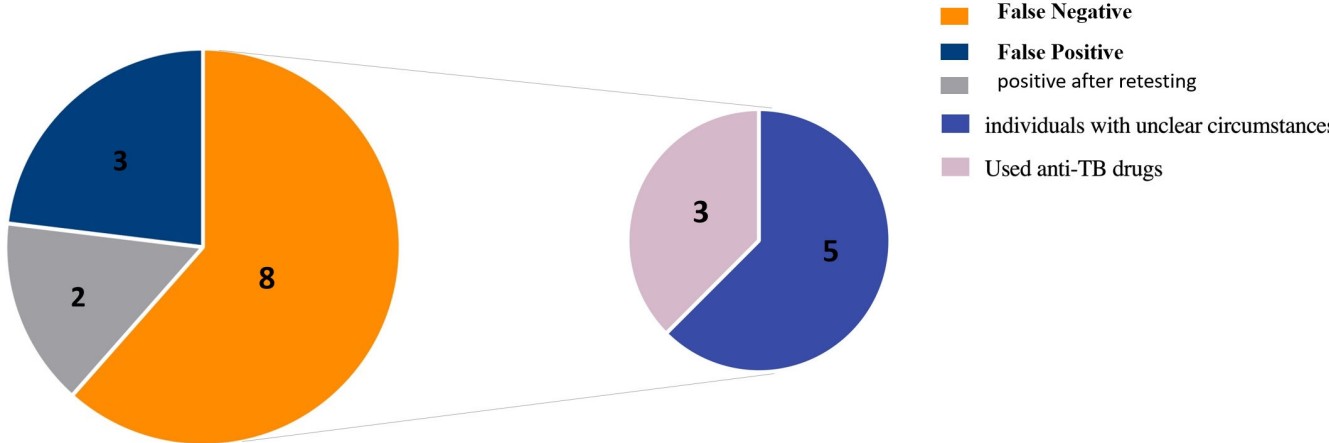

**FIG 3** Distribution of false-negative and false-positive results.

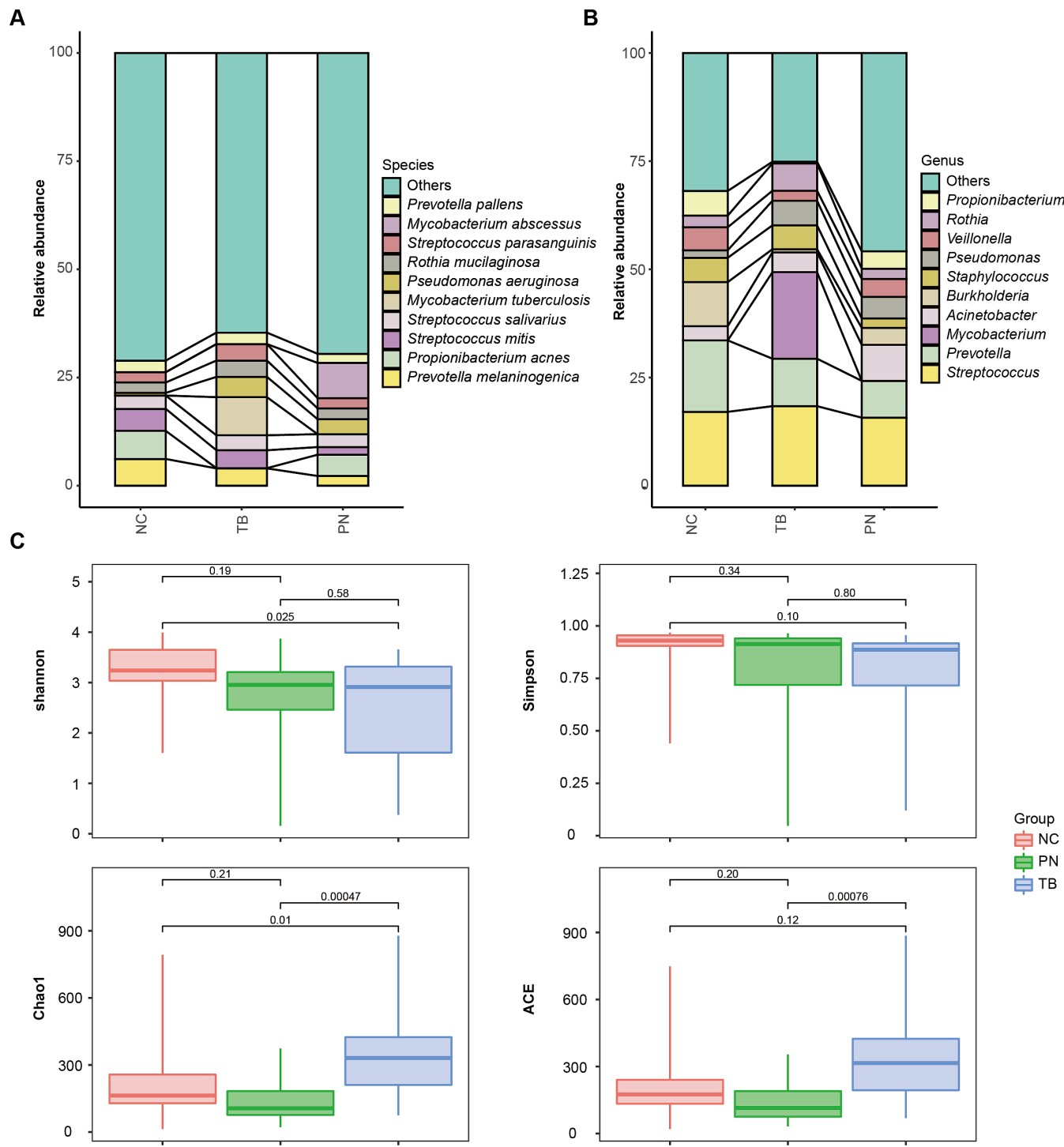

**FIG 4** Pulmonary microbiome analysis of three different groups, including NC group (no significant lung infection), TB group (tuberculosis-positive patients), PN group (tuberculosis-negative patients). (A) The average relative abundance of the top 10 species in the three groups. (B) The average relative abundance of the top 10 genus in the three groups. (C) The α-diversity of three different groups.

annually (23). Additionally, mNGS has a distinct advantage in detecting mixed infections with other pathogens and can also be applied to detect drug resistance in MTBC (24–26).

Research suggests that the majority of false-negative results stem from imperfect wet lab procedures, with microbial concentration being the most critical factor affecting these outcomes (21). MTBC possesses a unique and complex cell wall structure, rich

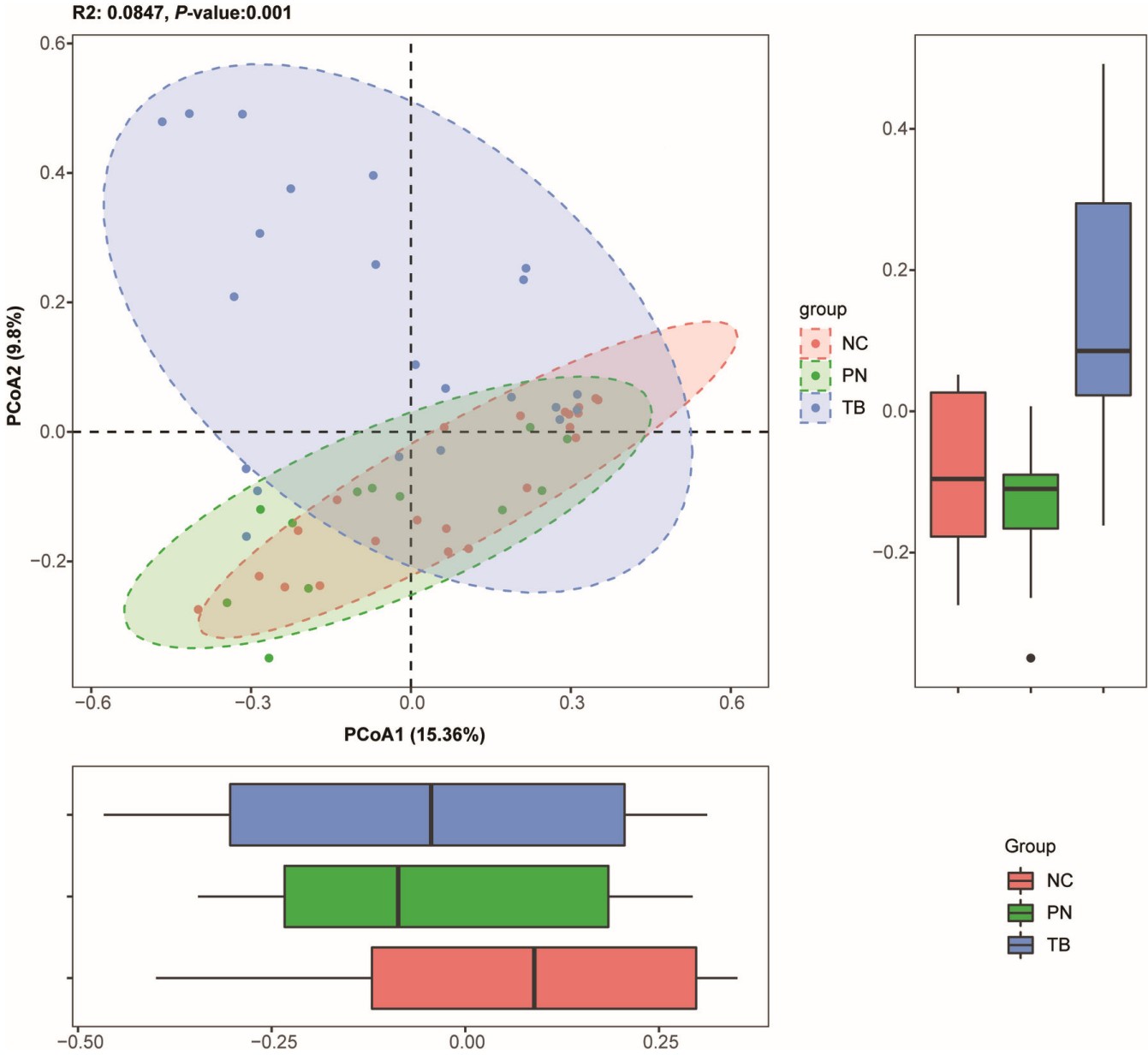

**FIG 5** The plot shows β-diversity among the three groups, with each point representing a sample and colors indicating different groups. Significant differences in β-diversity were found ($P < 0.05$). The bar charts show differences between groups along PCoA1 and PCoA2.

in lipids and polysaccharides, establishing a strong permeability barrier (22, 23). This complexity necessitates specialized methods to disrupt the cell wall for nucleic acid extraction, with the method of disruption being a primary determinant of extraction efficiency. In this study, the cell disruption process involved a shaking intensity of M/S 6.0, with 10 cycles, each lasting 1 min and 10 s, and a 10 s pause after each cycle. When analyzing results, samples diagnosed with MTBC infection but showing negative results were retested, increasing the disruption strength to M/S 7.0. Following retesting, two samples were found to contain MTBC, yielding a detection rate of 66.7% (2/3) and demonstrating that increased disruption strength aids in the detection of the bacterium. Therefore, when suspecting MTBC infection, it is necessary to incorporate or choose better cell disruption methods during the testing process. Additionally, clinicians should provide more clinical-related information at the time of submission, especially for special sample types such as cerebrospinal fluid, considering the possibility of MTBC infection

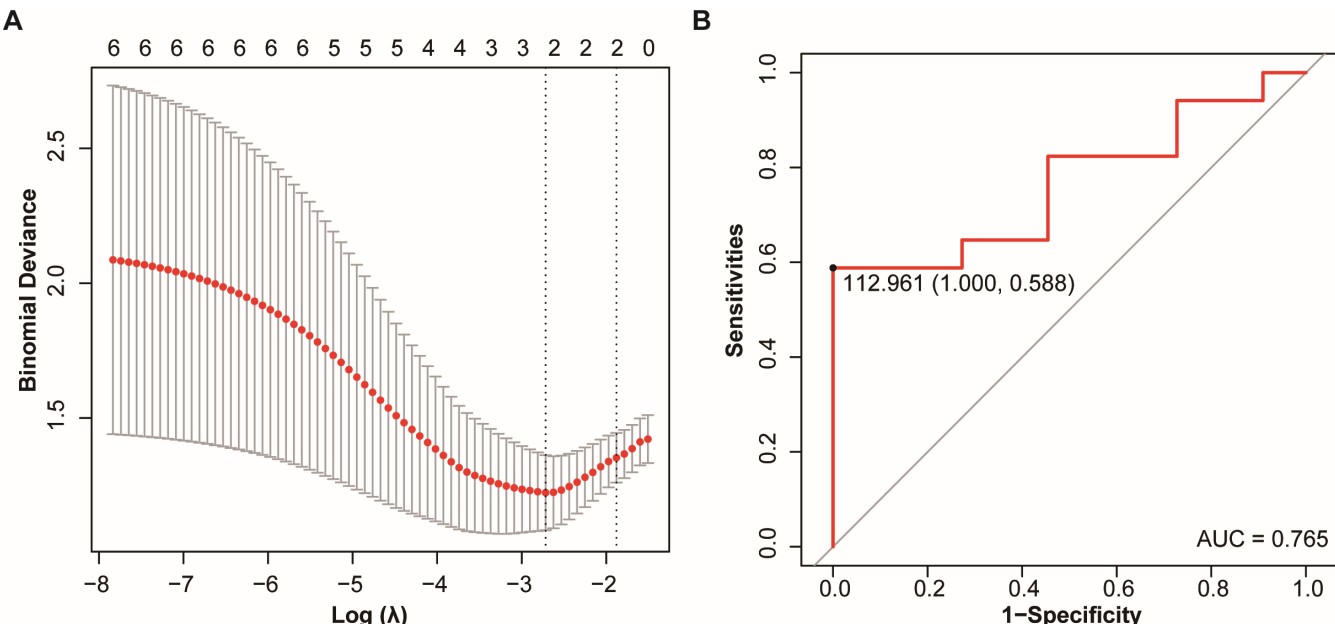

**FIG 6** (A) The Lasso analysis was conducted on *Streptococcus parasanguinis*, *Rothia mucilaginosa*, *Streptococcus mitis*, *Prevotella melaninogenica*, as well as the Shannon index and Chao1 index. The selected important feature from the analysis was Shannon index and Chao1 index. (B) ROC curve of *Y* for predicting MTBC infection. ROC curve, receiver operating characteristic curve.

to increase disruption strength and thereby improve detection rates. Besides MTB, most fungal cell walls also contain high-strength *β−1,3-glucans* and *α-1,3-glucans*, making them rigid (24) and similarly requiring appropriate disruption methods for nucleic acid extraction.

The presence of abundant human-origin nucleic acids in clinical samples is another factor leading to false-negative results (21, 25). In the mNGS testing workflow, de-hosting remains a critical step affecting the sensitivity of results. With large amounts of host-free nucleic acids present in most clinical samples and relatively lower abundance of microbial nucleic acids, high host rates can lead to sequencing outcomes predominantly originating from the human genome, reaching up to 91.6% (26). This may result in pathogen undetection and false-negative outcomes.

Moreover, the use of clinical anti-TB drugs can also impact testing outcomes. Studies have shown that when anti-TB drugs are used for more than 3 months, the rate of MTBC detection through mNGS significantly decreases compared to cases without exposure to such drugs (27). In this study, analysis suggests that pre-submission use of effective anti-TB drugs could lower pathogen levels in the body, leading to negative mNGS results. Furthermore, four samples tested negative for all tests, which we attribute to potential issues related to sample collection quality and collection sites.

We speculate potential sources of positivity, including contamination from both wet and dry lab processes. Wet lab steps involving sample collection and preparation could introduce contamination from lab reagents, consumables, or the environment.

**TABLE 3** Comparison of *Y* values with the cut-off value for predicting tuberculosis in five cases of unknown False-negative results

| Sample ID | Prediction | Cut-off value | Chao1 | Shannon | *Y* value |
|-----------|------------|---------------|--------|---------|-----------|
| 18 | Positive | 112.96 | 402.07 | 3.65 | 125.83 |
| 46 | Negative | | 230.11 | 3.75 | 72.01 |
| 87 | Negative | | 307 | 3.8 | 96.08 |
| 96 | Negative | | 317.66 | 4.19 | 99.41 |
| 162 | Positive | | 433.72 | 3.57 | 135.74 |

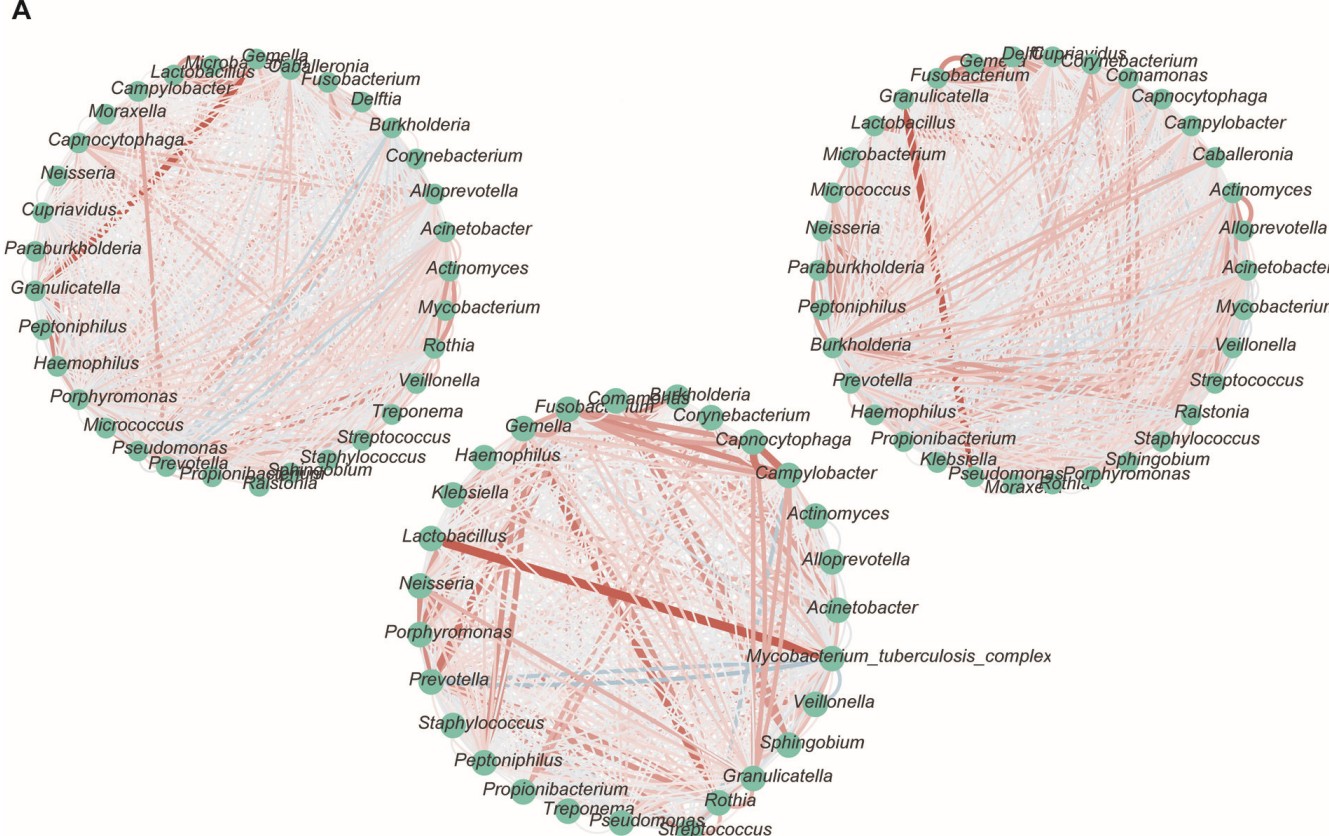

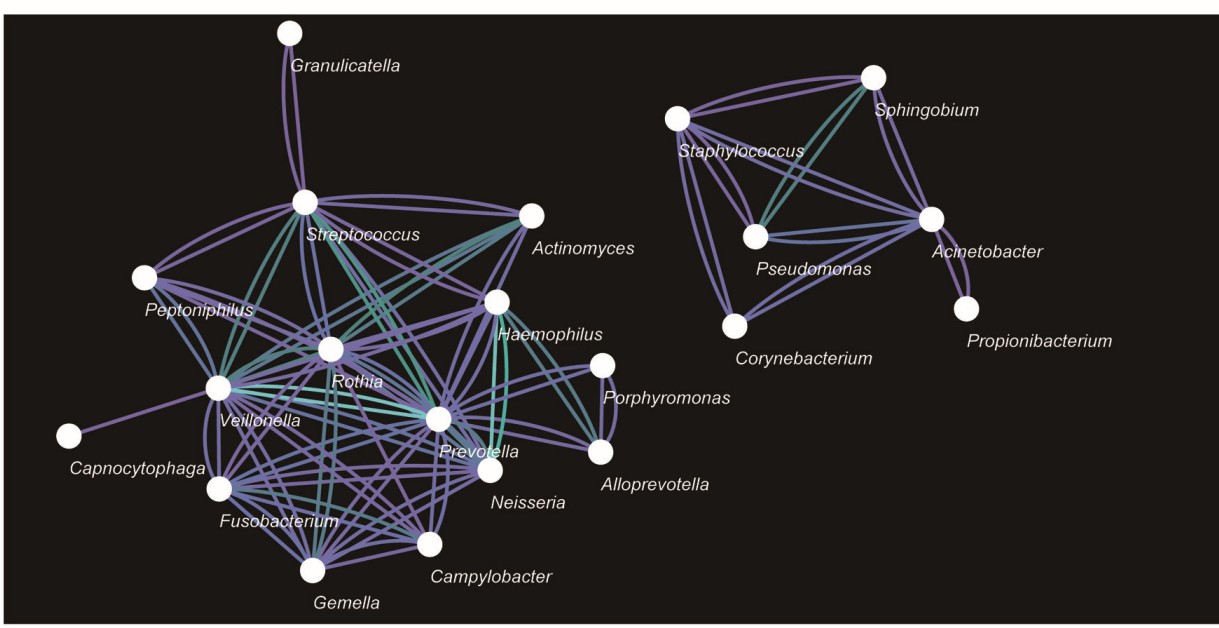

FIG 7  (A) From left to right are the microbiome network diagrams for the NC, TB, and PN groups. Nodes represent microbial genera, and edges represent correlations between genera. Red edges indicate positive correlations, while blue edges indicate negative correlations. The darker the color and the thicker the edge, the stronger the correlation. (B) The consensus network of the NC, TB, and PN groups. In the network, the bluer the edge color, the more conserved the relationship is within the network.

Establishing a background microbial database for long-term monitoring in our lab is suggested (21). Additionally, cross-contamination in the lab, especially during the

preparation of low- and high-concentration samples, and sequencer contamination, is also significant concerns. In an analysis of a false-positive result, we identified that using the same index (C316D) for a sample that had previously detected strong positive specimens could leave residual nucleic acid fragments in the sequencer pipelines. If the same pathogen is detected multiple times within a period using the same index, pipeline residue should be considered. In such cases, adding PCR verification to reports is recommended to clarify if TB detection is false-positive. Moreover, labs should increase the cleaning frequency of sequencing equipment pipelines to quickly eliminate contamination and use the index for negative quality control before reuse to ensure any contamination has been cleared.

During the experiment, due to the genomic similarities among microorganisms, classification may erroneously categorize sequences as the wrong species within the same genus, leading to false positives in the experimental results. Special attention should be paid to short sequences with repetitive structures (palindromes), as these may increase the likelihood of false positives. In this study, two patients were found to have MTBC sequences with lengths of 38 bp and 36 bp, respectively. After analysis and considering clinical symptoms, these results were determined to be false positives. Therefore, we believe that there are potentially similar genomic fragments between different species, and the analysis system might make mistakes leading to false positives. It is necessary for laboratories to establish corresponding SOP documents when interpreting reports, taking clinical information into account as well as the detection conditions of the sequences, especially when the list of detected species contains strongly positive ones, due to the nonspecific comparison between TB and other species leading to false positives.

Currently, it has been found that different pulmonary diseases can cause changes in lung microecology (28–30). Infection with MTBC can lead to changes in the lung microbiome. This study found that after infection with TB, MTBC became the dominant microorganism in the lung microbiome, whereas in people without lung infections, the main dominant microorganisms were *Veillonella atypica* and *Neisseria subflava*. The NC group and the PN group had similar α-diversity, but there was a significant difference from the TB group, partially similar to the results of other studies (10) and different from some (9, 31), possibly affected by factors such as the number of samples and whether anti-TB medication was used (32). There was a significant difference in β-diversity between different groups, similar to the findings of Anthony M. Cadena's study, which found that macaques had an increase in lung microbiota diversity 1 month after MTBC infection (33). We consider that a major reason for this change is the inflammatory response caused by MTBC infection altering vascular permeability and the damage to epithelial cells making them more conducive to bacterial adhesion (34). Under healthy conditions, relatively fewer microbes are present due to respiratory mucus, alveolar surfactants, and host immune defenses (35). The changes in Shannon and chao1 indices may be closely associated with significant alterations in the lung microbiota following MTBC infection. This study demonstrated that the Shannon and chao1 indices have potential utility in predicting TB, with AUC values of 0.765, indicating moderate predictive capability. This approach not only enhances the accuracy of TB detection but also provides new insights into the potential impact of MTBC infection on the lung microbiota.

In co-occurrence network analysis, we observed variations in microbial interactions. During MTBC infection, there was a notable enhancement in positive correlations among microorganisms, indicating increased synergy among them following invasion by MTBC. It is suggested in previous studies that the microbial communities in the lungs play a protective role in the host. Healthy microbial populations can prevent the invasion of potential pathogens through mechanisms such as generating colonization resistance, competing for nutrients, or producing bacteriocins to kill competing microorganisms (36, 37). Our study proposes that the augmented synergy among microbial communities following MTBC infection may represent a protective mechanism by which microbes

fortify the host against pathogen colonization. Moreover, through network analysis, we identified nine core microbial taxa commonly present across the three groups, including *Streptococcus, Veillonella, Rothia, Actinomyces, Haemophilus, Prevotella, Alloprevotella, Neisseria, Fusobacterium,* consistent with findings by Fernanda Valdez-Palomares and others (9). These nine taxa at the network core are considered conservative microbial communities less susceptible to MTBC infection. It has been established that certain taxa within this core, such as *Prevotella* and *Veillonella* genera, are capable of producing short-chain fatty acids (SCFAs), enhancing immune responses, and suppressing inflammation on mucosal surfaces (38, 39). However, the roles of other core microbial taxa in relation to MTBC warrant further investigation. Conversely, non-core microbial taxa are more susceptible to influences from the host and external environment and may undergo changes with factors such as infection and administration of anti-TB medications.

## Conclusion

mNGS, as an emerging detection technology, holds unique advantages over traditional diagnostic methods in the diagnosis of TB, providing a basis for the early diagnosis and treatment of clinical TB. This study analyzes factors that may lead to false-negative and false-positive results with this method and emphasizes the continuous optimization of experimental methods to ensure the accuracy of results. Additionally, we applied mNGS to analyze the lung microecology of patients under different conditions and used the assessment index $Y$ to assist in the diagnosis of MTBC infection. Co-occurrence network analysis revealed changes in the interactions among microorganisms. This study has certain limitations, as BALF is an invasive detection method, preventing the collection of samples from healthy individuals; therefore, patients with pulmonary shadows were collected as a control group. Moreover, the study lacks longitudinal research to analyze changes in the microecology of the same patient in different states, such as healthy, infected with TB, and post-treatment, which will be a focus of subsequent research.

## ACKNOWLEDGMENTS

This work was supported by the National Natural Science Foundation of China (82272380, 82072318), National High Level Hospital Clinical Research Funding (2022-PUMCH-B-028, 2022-PUMCH-C-060, 2022-PUMCH-B-074), the National Key Research and Development Program of China (2021YFC2301002), National Key Research and Development Program of China (2022YFC2603800), Beijing Municipal Science & Technology Commission (No. Z221100007422076) and supported by the Non-profit Central Research Institute Fund of Chinese Academy of Medical Sciences (2023-PT330-01). The funders had no role in the study design, data collection and analysis, decision to publish, or preparation of the manuscript.

## AUTHOR AFFILIATIONS

[1]Department of Clinical Laboratory, State Key Laboratory of Complex Severe and Rare Diseases, Peking Union Medical College Hospital, Chinese Academy of Medical Sciences and Peking Union Medical College, Beijing, China

[2]Department of Clinical Laboratory, The second Affiliated Hospital of Zunyi Medical University, Zunyi, Guizhou, China

[3]Key Laboratory of Pathogen Infection Prevention and Control (Peking Union Medical College), Ministry of Education, Beijing, China

[4]Department of Scientific Affairs, Hugobiotech Co., Ltd., Beijing, China

[5]Department of Clinical Laboratory, The Affiliated Qingdao Third People's Hospital of Qingdao University, Qingdao, Shandong, China

## AUTHOR ORCIDs

Dong Zhang http://orcid.org/0009-0002-6638-728X
Ying Zhao https://orcid.org/0000-0002-7093-1121
Yingchun Xu http://orcid.org/0000-0002-7126-9459
Xiaojun Ge http://orcid.org/0000-0001-9846-365X
Qiwen Yang http://orcid.org/0000-0001-7272-3900

## DATA AVAILABILITY

The data that support the findings of this study are available in Figshare at the following DOI: https://doi.org/10.6084/m9.figshare.28060463.

## ADDITIONAL FILES

The following material is available online.

### Supplemental Material

**Supplemental figures (Spectrum01563-24-s0001.docx).** Figures S1 to S3.

### Open Peer Review

**PEER REVIEW HISTORY (review-history.pdf).** An accounting of the reviewer comments and feedback.

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
