## [Reviewer comments · Microbiology Spectrum]

Microbiology Spectrum

Integrative study of pulmonary microbiome and clinical diagnosis in Pulmonary Tuberculosis Patients

hongli sun, Qiuyue Chen, Dong Zhang, Min-Ya Lu, Yao Wang, Huiting Su, Yi Gao, Jiayu Guo, Ying Zhao, Juan Du, Cun Liu, Ying-Chun Xu, Xiaojun Ge, and Qiwen Yang

Corresponding Author(s): Qiwen Yang, Peking Union Medical College Hospital

Review Timeline:

Submission Date:	July 24, 2024
Editorial Decision:	November 8, 2024
Revision Received:	January 6, 2025
Accepted:	April 19, 2025

Editor: Fei Chen

Reviewer(s): Disclosure of reviewer identity is with reference to reviewer comments included in decision letter(s). The following individuals involved in review of your submission have agreed to reveal their identity: Tingting Yang (Reviewer #1)

Transaction Report:

DOI: <https://doi.org/10.1128/spectrum.01563-24>

Re: Spectrum01563-24 (Integrative study of pulmonary microbiome and clinical outcomes in Pulmonary Tuberculosis Patients)

Dear Dr. Qiwen Yang:

Thank you for the privilege of reviewing your work. Below you will find my comments, instructions from the Spectrum editorial office, and the reviewer comments.

Revision Guidelines

Sincerely,
Fei Chen
Editor
Microbiology Spectrum

Reviewer #1 (Comments for the Author):

In this study, Sun and colleagues aimed to assess the diagnostic accuracy of metagenomic next-generation sequencing (mNGS) for the detection of *Mycobacterium tuberculosis*. They conducted an analysis of 236 bronchoalveolar lavage fluid samples and examined how variations in the pulmonary microbiome can aid in tuberculosis diagnosis by comparing the pulmonary microbial communities among samples from patients with tuberculosis (TB group), clinically outdated TB (PN group), and no significant lung infection (NC group). The sensitivity of mNGS was found to be 73.33%, which was higher than that of Xpert, culture, RT-PCR, and sputum smear tests. However, there were three false positive results, leading to a specificity of 98.54%. In

comparison to the NC group, the TB group exhibited significantly decreased alpha diversity, with an area under the curve (AUC) of 0.711 as assessed using the Shannon index.

I have a few major concerns with this manuscript:

1. Several crucial results were erroneously placed in the Discussion section, including the exploration of the reasons for false positive and false negative results, and use of the Shannon index to help rule out false positive results. These results are the highlights of the article and form the primary basis of the Discussion section. The corresponding figures and tables should also be displayed.
2. The discussion section is excessively verbose and requires conciseness.
3. Lack of "Data availability" and "Acknowledgements".

Detailed comments:

1. The significance of Table 1 in the paper seems minimal, suggesting that moving it to the Supplementary section.
2. Please provide the specific numerical values for positive and negative predictions of each method in Table 2.
3. Some content in the Discussion repeats information from the Introduction.
4. Lack of statistical methods.
5. The authors mention that "In this study, one patient was detected with a short and simple sequence of *Mycobacterium tuberculosis*, which, despite its specificity, was still a false positive". Is it possible that this patient was actually a true positive?
6. In "2.5 Analysis of Microbial Community Structure", references to the vegan package and the ape package are missing.
7. The abbreviation should be introduced when it first appears in the text and then used consistently throughout. Please review and revise.
8. "*Mycobacterium tuberculosis*" should be italicized.
9. Lack of Line numbers.
10. The formatting is inconsistent, with some paragraphs indented at the first line and others not.

Reviewer #2 (Comments for the Author):

The manuscript by Sun Hongli et al. presents insights into the use of mNGS in tuberculosis diagnosis and the profile of the pulmonary microbiome. The study aims to evaluate the diagnostic performance of metagenomic next-generation sequencing (mNGS) in detecting *Mycobacterium tuberculosis* and to analyze the pulmonary microbiome in tuberculosis patients to improve diagnostic precision. Several major weaknesses related to methodology and overall organization should be addressed.

Major comments:

1. Some methods are not clearly described in certain sections, for example, AUROC is not mentioned in the article. Some bioinformatic information is missing, for example, how many reads were obtained, how the raw sequencing data was processed, and which software was used to obtain the OTU table.
2. The intervention was not well defined in this paper. The title of the article is "Integrative study of pulmonary microbiome and clinical outcomes in Pulmonary Tuberculosis Patients," but the abstract states, "This study aimed to explore the diagnostic performance of metagenomic next-generation sequencing (mNGS) for the detection of *Mycobacterium tuberculosis*, and to enhance its diagnostic accuracy through the analysis of pulmonary microbiome mNGS data." Please consider making the key content consistent.
3. In the results section, this work more focused on the ability of a test or method to detect a specific target, *Mycobacterium tuberculosis*. Research motive where the hypothesis of this study is not clear in the paper. How the microbiome analysis contribute or link to the detection of *Mycobacterium tuberculosis*.
4. Using the Shannon index to predict tuberculosis bacteria is confusing. What is the practical application value? How about bacterial species?
5. Figures: the numbers are not fully displayed (Fig.4B) and the font size is too small (Fig. 4C Lefse).

In this study, Sun and colleagues aimed to assess the diagnostic accuracy of metagenomic next-generation sequencing (mNGS) for the detection of *Mycobacterium tuberculosis*. They conducted an analysis of 236 bronchoalveolar lavage fluid samples and examined how variations in the pulmonary microbiome can aid in tuberculosis diagnosis by comparing the pulmonary microbial communities among samples from patients with tuberculosis (TB group), clinically outdated TB (PN group), and no significant lung infection (NC group). The sensitivity of mNGS was found to be 73.33%, which was higher than that of Xpert, culture, RT-PCR, and sputum smear tests. However, there were three false positive results, leading to a specificity of 98.54%. In comparison to the NC group, the TB group exhibited significantly decreased alpha diversity, with an area under the curve (AUC) of 0.711 as assessed using the Shannon index.

I have a few major concerns with this manuscript:

1. Several crucial results were erroneously placed in the Discussion section, including the exploration of the reasons for false positive and false negative results, and use of the Shannon index to help rule out false positive results. These results are the highlights of the article and form the primary basis of the Discussion section. The corresponding figures and tables should also be displayed.
2. The discussion section is excessively verbose and requires conciseness.
3. Lack of “Data availability” and “Acknowledgements”.

Detailed comments:

1. The significance of Table 1 in the paper seems minimal, suggesting that moving it to the Supplementary section.
2. Please provide the specific numerical values for positive and negative predictions of each method in Table 2.
3. Some content in the Discussion repeats information from the Introduction.
4. Lack of statistical methods.
5. The authors mention that “In this study, one patient was detected with a short and simple sequence of *Mycobacterium tuberculosis*, which, despite its specificity, was still a false positive”. Is it possible that this patient was actually a true positive?
6. In “2.5 Analysis of Microbial Community Structure”, references to the vegan package and the ape package are missing.
7. The abbreviation should be introduced when it first appears in the text and then used consistently throughout. Please review and revise.
8. “*Mycobacterium tuberculosis*” should be italicized.
9. Lack of Line numbers.
10. The formatting is inconsistent, with some paragraphs indented at the first line and others not.

Response to Reviewers

We sincerely appreciate the time and effort you have invested in reviewing our manuscript titled
“Integrative study of pulmonary microbiome and clinical outcomes in Pulmonary Tuberculosis
Patients” (Spectrum 01563-24). We are grateful for your valuable comments and constructive
feedback, which have helped us to significantly improve the quality of our work. In this revised
version, we have carefully addressed all the comments and suggestions provided by the reviewers.
Below, we provide a detailed, point-by-point response to each of the reviewers' comments, We
hope that these revisions meet your expectations and address all of the concerns raised.

**Response to Reviewer #1**

**Responses to Major Concerns**

**Major Concerns 1:** *Several crucial results were erroneously placed in the Discussion section,*
*including the exploration of the reasons for false positive and false negative results, and use of the*
*Shannon index to help rule out false positive results. These results are the highlights of the article*
*and form the primary basis of the Discussion section. The corresponding figures and tables should*
*also be displayed.*

**Author Response:** We appreciate the valuable comments provided by the reviewer. We realize
that some key results, such as the causes of false positives and false negatives, as well as the
application of the Shannon index, should indeed be more appropriately placed in the Results
section. We have moved these results to the Results section and have accordingly updated the
figures and tables to present the data more clearly. Thank you for your suggestion, and we believe
these revisions significantly improve the logical flow and structure of the manuscript.

The detailed revisions are as follows:

1. Regarding the issues of false negatives and false positives, we have added the following content
in the Results section: In the mNGS detection, eight false-negative cases were observed. Among
these, three cases involved individuals who had undergone anti-TB treatment. The remaining five
individuals have unclear circumstances. Additionally, three false-positive cases were detected.
Furthermore, two samples initially tested negative but yielded positive results upon secondary
testing following the application of specialized experimental procedures. The detailed
classification of these findings is presented in Figure 3. (see revised manuscript RESULTS, page
17, lines 295-302).

2. Regarding the issues of application of the Shannon index, We provided a detailed description of
the selection and application of the indices in the results section. The specific modifications are as
follows: “we identified factors that showed significant differences between the TB group and NC
group, with a focus on the following microbial species: *Streptococcus parasanguinis*, *Rothia*
*mucilaginosus*, *Streptococcus mitis*, *Prevotella melaninogenica*, as well as the Shannon index and
Chao1 index. Through Lasso regression analysis, we selected two factors that were significantly
associated with clinical diagnosis of TB: the Shannon index and Chao1 index. Based on these
significant factors, we constructed the following assessment index:

$$39 Y = -0.004 \times \text{Shannon index} + 0.313 \times \text{Chao1 index}$$

We then used this evaluation index to plot the ROC curve. The area under the ROC curve (AUC)
was 0.765, indicating that the model has a moderate ability to discriminate between groups. Based
on the ROC curve, we determined the cut-off value to be 112.96, which serves as the threshold for
clinical diagnosis of TB, as shown in Figure 6. Using this cut-off value, we analyzed the above
five false-negative cases of unknown cause and identified two cases of tuberculosis infection, the

details are shown in table 3.”(see revised manuscript RESULTS, page 18, lines 321-337)

**Major Concerns 2:** *The discussion section is excessively verbose and requires conciseness.*

**Author Response:** Thank you for your valuable feedback. In response to your comment regarding

the discussion section being excessively verbose, we have revised the manuscript by removing

redundant content that overlaps with the introduction. Additionally, we have streamlined the

methodology comparison to focus more on the key findings. We believe these changes have

helped to enhance the clarity and conciseness of the discussion while maintaining the essential

information. After the revisions, the word count of the discussion section has been reduced from

1,924 to 1,373. The revised discussion focuses primarily on the causes of false-negative and

false-positive results, as well as the analysis of microbial ecological changes.(see revised

manuscript DISCUSSION, page 21-22, lines 362-580)

**Major Concerns 3:** *Lack of "Data availability" and "Acknowledgements."*

**Author Response:** Thank you for pointing out this omission. We have added a "Data Availability"

statement and an "Acknowledgements" section in the revised manuscript. The details are as

follows:

3.1 Data Availability Statement: We have included a statement regarding the availability of the

data used in this study. The text is as follows:

“The data that support the findings of this study are available from DOI:

10.6084/m9.figshare.28060463. ”(see revised manuscript, page 36, lines 614-616)

3.2 Acknowledgements Section: We have added an Acknowledgements section to express our

gratitude to those who contributed to the study. The text is as follows:
The author(s) declare financial support was received for the research, authorship, and/or
publication of this article. This work was supported by the National Natural Science Foundation of
China (82272380, 82072318), National High Level Hospital Clinical Research Funding
(2022-PUMCH-B-028,2022-PUMCH-C-060, 2022-PUMCH-B-074), the National Key Research
and Development Program of China (2021YFC2301002), and supported by the Non-profit Central
Research Institute Fund of Chinese Academy of Medical Sciences(2023-PT330-01). The funders
had no role in the study design, data collection and analysis, decision to publish, or preparation of
the manuscript.(see revised manuscript Acknowledgements, page 35, lines 602-612).

**Responses to Detailed Comments**

**Reviewer's Detailed Comment 1:** *The significance of Table 1 in the paper seems minimal,*
*suggesting that moving it to the Supplementary section.*

**Author Response:** Thank you for your valuable feedback on Table 1. We appreciate your
suggestion to move it to the supplementary section. However, after careful consideration, we
believe that Table 1 provides critical clinical information that is integral to understanding the
characteristics of the study population.

To address your feedback, we have enhanced Table 1 by adding key clinical characteristics,
specifically History of TB (individuals), NTM (individuals), and Deaths (individuals). These
additional factors provide essential context for the study population and are crucial for
understanding the clinical profiles of the patients included in this study. We believe that these
updates make the table more informative and relevant for readers.

Given the central role this table plays in presenting important baseline data, we respectfully
request to retain it in the main text. We believe that including this table in the main manuscript
will ensure that readers can easily access this critical information without having to refer to the
supplementary section.

We hope that these modifications address your concerns, and thank you again for your
constructive comments. (see revised manuscript, table1).

**Reviewer's Detailed Comment 2:** *Please provide the specific numerical values for positive and*
*negative predictions of each method in Table 2.*

**Author Response:** We appreciate the reviewer's suggestion. We have now included the specific
numerical values for positive and negative predictions of each method in the Supplementary
Materials (see Supplementary Fig S2). Additionally, we have updated the main manuscript to
reference this new supplementary table (see revised text on page16, line 287-288).

**Reviewer's Detailed Comment 3:** *Some content in the Discussion repeats information from the*
*Introduction.*

**Author Response:** We appreciate the reviewer's insightful comment. To address this issue, we
have streamlined the Discussion section by removing the first and second paragraphs, which
previously included background information on the limitations of existing TB diagnostic methods.
The revised Discussion now directly focuses on the interpretation of our study findings. We
believe this revision has effectively eliminated the redundancy and improved the overall clarity of
the manuscript (see revised manuscript DISCUSSION, page 21-22, lines 362-384).

**Reviewer's Detailed Comment 4:** *Lack of statistical methods.*

**Author Response:** Thank you for pointing out the lack of statistical methods description in the
original manuscript. We appreciate your suggestion. In response, we have added a detailed
description of the statistical analysis methods used in this study in the Methods section.

Specifically, we have included:

- 1. Stated that categorical variables were analyzed using the chi-square test or Fisher's exact test,
based on the sample characteristics.
- 2. Explained that continuous variables were assessed for normality using the Shapiro-Wilk test.
- 3. Included details about the use of LASSO logistic regression for variable selection and ROC
curve analysis for evaluating predictive performance.

These additions aim to provide a comprehensive overview of the statistical analyses performed
and ensure the clarity and reproducibility of our results. We have updated the manuscript
accordingly (see the revised Statistical Analysis , page14, lines 243-260).

**Reviewer's Detailed Comment 5:** *The authors mention that "In this study, one patient was*
*detected with a short and simple sequence of Mycobacterium tuberculosis, which, despite its*
*specificity, was still a false positive". Is it possible that this patient was actually a true positive?*

**Author Response:** Thank you for your valuable feedback. Based on the sequence length and
simplicity, other test results, and clinical diagnosis, we concluded that these sequences were false
positives. We apologize for the confusion caused by presenting cases of the same issue separately
in the original version. In the revised manuscript, I have integrated the analysis of these issues (see
the revised Discussion , page29, lines 499-502). We hope this clarification will address your
concerns, and we appreciate your feedback, which will help improve the accuracy of our

manuscript.

**Reviewer's Detailed Comment 6:** *In "2.5 Analysis of Microbial Community Structure",*

*references to the vegan package and the ape package are missing.*

**Author Response:** Thank you for pointing this out. We have carefully reviewed the manuscript

and added the appropriate references for both the vegan package and the ape package in the

revised version. Specifically:

For the vegan package, we have cited:

Oksanen J, Kindt R, Legendre P, O'Hara B, Stevens MHH, Oksanen MJ. 2007. The vegan

package. *Community ecology* package 10:631-637.(see the revised Method, page11, lines 204)

For the ape package, we have cited:

Paradis E, Schliep K. 2019. ape 5.0: an environment for modern phylogenetics and evolutionary

analyses in R. *Bioinformatics* 35:526-528.(see the revised Method, page11, lines 209)

These references have been included in the main text (Section 2.5) and added to the reference list.

**Reviewer's Detailed Comment 7:** *The abbreviation should be introduced when it first appears in*

*the text and then used consistently throughout. Please review and revise.*

**Author Response:** Thank you for your helpful suggestion. We have reviewed the manuscript and

made the necessary revisions to ensure that all abbreviations are introduced when they first appear

in the text and then used consistently throughout. Specifically, we have introduced the following

abbreviations as requested:

Metagenomic next-generation sequencing (mNGS), *Mycobacterium tuberculosis* (*M. tuberculosis*),

Bronchoalveolar lavage fluid (BALF), Tuberculosis (TB), World Health Organization (WHO)

Xpert MTB/RIF (Xpert), Receiver operating characteristic (ROC), Least Absolute Shrinkage and

Selection Operator (LASSO), Area under the curve (AUC)

We hope these revisions address your concerns and improve the clarity and consistency of the
manuscript. Thank you again for your valuable feedback.

**Reviewer's Detailed Comment 8:** *"Mycobacterium tuberculosis" should be italicized.*

**Author Response:** Thank you for your helpful comment. We have revised the manuscript to
ensure that "Mycobacterium tuberculosis" is italicized throughout, as well as other terms that
require italicization, including *non-tuberculous mycobacteria*, *β -1,3-glucans*, and *α -1,3-glucans*.

We hope this addresses your concern and improves the consistency and accuracy of the
manuscript. Thank you again for your valuable feedback.

**Reviewer's Detailed Comment 9:** *Lack of Line numbers.*

**Author Response:** Thank you for pointing out the absence of line numbers in the original
submission. We apologize for the oversight. As requested, we have now added line numbers
throughout the revised manuscript to facilitate easier reference and review.

**Reviewer's Detailed Comment 10:** *The formatting is inconsistent, with some paragraphs*
*indented at the first line and others not.*

**Author Response:** Thank you for your constructive feedback. I have carefully reviewed the
formatting of the manuscript and made the necessary adjustments to ensure consistent indentation
for all paragraphs.

**Response to Reviewer #2**

**Comment 1:** *Some methods are not clearly described in certain sections, for example, AUROC is*
*not mentioned in the article. Some bioinformatic information is missing, for example, how many*

reads were obtained, how the raw sequencing data was processed, and which software was used to
obtain the OTU table.

**Author Response:** Thank you for your valuable feedback. We have addressed the issues you
raised as follows:

1. AUROC Not Mentioned: We have added relevant explanations of AUROC in the Statistical
analysis(see revised Statistical analysis, page15, lines 256--260) and results sections to ensure it is
fully explained in the manuscript. For details, please refer to the revised version of the
manuscript.(see revised Result, page19, lines 330-334).

2. Details on Bioinformatic information: I have added bioinformatic information in the revised
manuscript.(see revised Method, pag11, lines 189-200)

3. about OTU table: Thank you for your insightful comment. Our study is based on metagenomic
sequencing rather than amplicon-based sequencing (e.g., 16S rRNA), and thus we did not generate
an OTU table. Our research data can be accessed at DOI: 10.6084/m9.figshare.28060463.

**Comment 2:** *The intervention was not well defined in this paper. The title of the article is*
*"Integrative study of pulmonary microbiome and clinical outcomes in Pulmonary Tuberculosis*
*Patients," but the abstract states, "This study aimed to explore the diagnostic performance of*
*metagenomic next-generation sequencing (mNGS) for the detection of Mycobacterium*
*tuberculosis, and to enhance its diagnostic accuracy through the analysis of pulmonary*
*microbiome mNGS data." Please consider making the key content consistent.*

**Author Response:** Thank you for your detailed review and valuable feedback on our paper.
Regarding the concern that the intervention was not well defined, we have revised the title and
research objectives to ensure better consistency. We believe that this revision ensures better

alignment between the research objectives and the article title, and more clearly conveys the core
focus of the study.

Revised title and research objective:

title: Integrative study of pulmonary microbiome and clinical diagnosis in Pulmonary Tuberculosis

Patients

research objective: This study investigated the diagnostic potential of metagenomic

next-generation sequencing(mNGS) for detecting *Mycobacterium tuberculosis*(*M. tuberculosis*) in

pulmonary tuberculosis patients. We analyzed pulmonary microbiome data to assess its impact on

mNGS diagnostic accuracy and explored the association between microbiome profiles and clinical

diagnosis. (see revised Abstract, page1, lines 5-9).

**Comment 3:** *In the results section, this work more focused on the ability of a test or method to*

*detect a specific target, *Mycobacterium tuberculosis*. Research motive where the hypothesis of this*

*study is not clear in the paper. How the microbiome analysis contribute or link to the detection of*

**Mycobacterium tuberculosis*.*

**Author Response:** We appreciate the reviewer's comments regarding the clarity of the study's

motive and hypothesis. Based on your suggestions, we have clarified the research motive in the

manuscript. Below is a detailed explanation in response to your concerns.

1. Clarifying the research motive and hypothesis: The aim of this study was to evaluate the

diagnostic performance of the mNGS method for detecting *M. tuberculosis* and to investigate

strategies for reducing false-positive and false-negative results. To further enhance the sensitivity

of the detection, we conducted a microbiome analysis to compare the microbial community

composition between healthy individuals and tuberculosis patients, exploring whether these

microbial features could serve as auxiliary biomarkers to improve the accuracy of *M. tuberculosis*
detection.(see revised INTRODUCTION, page6, lines 95-102).

2. Explaining the contribution of microbiome analysis: While comparing the performance of
different diagnostic methods for *Mycobacterium tuberculosis* detection, we also observed a high
rate of false-negative results in the mNGS findings. This observation prompted us to further
investigate whether microbiome analysis could provide additional insights. The microbiome
analysis revealed significant ecological differences between healthy individuals and tuberculosis
patients, which may offer extra diagnostic value. In this study, we used indicators identified
through microbiome analysis—specifically, the Shannon index, Chao1 index, and differential
microbial features—to perform LASSO regression analysis, selecting the Shannon index and
Chao1 index as key features to establish the diagnostic model. Subsequent ROC analysis
demonstrated that these specific microbiome characteristics may serve as supplementary
biomarkers, potentially improving the specificity of MTB detection when used in combination
with mNGS.

**Comment 4:** *Using the Shannon index to predict tuberculosis bacteria is confusing. What is the*
*practical application value? How about bacterial species?*

**Author Response:** Thank you for your valuable feedback. We understand that using the Shannon
index to predict *Mycobacterium tuberculosis* might be confusing, as the Shannon index is
primarily a measure of microbial diversity rather than a direct indicator of a specific pathogen.
Here is the specific process we used for selection: We performed microbiome analysis comparing
the tuberculosis group and the healthy control group, identifying microbiota with significant
differences, including *Streptococcus parasanguinis*, *Rothia mucilaginosa*, *Streptococcus mitis*, and

*Prevotella melaninogenica*, as well as the statistically significant Shannon index and chao1.
Through LASSO analysis, the most important feature selected was the Shannon index and chao1,
then used to construct an ROC curve. We have added images and explanatory notes in the Results
section of the revised manuscript. (see revised RESULTS, page18-19, lines 304-320). Here, we
present the ROC curves for *Streptococcus parasanguinis*, *Rothia mucilaginosa*, and *Streptococcus*
*mitis*, analyzed individually. The corresponding figure is shown below.

**Comment 5:** *Figures: the numbers are not fully displayed (Fig. 4B) and the font size is too small*
*(Fig. 4C Lefse).*

**Author Response:** Thank you for your valuable feedback regarding the figures. We have carefully
addressed the issues you raised:

1. For Figure 4B: We have revised the figure to ensure that all numbers are fully displayed. The
updated figure has been included in the revised manuscript.

2. For the original Figure 4C (LEfSe): Considering your suggestion about the font size and the
overall relevance to the main text, we have modified the figure to improve readability and clarity.

Furthermore, to better align with the flow of the manuscript, we have moved this figure to the
supplementary materials section and renamed it as Supplementary Figure S3.

We hope these changes meet your expectations and enhance the quality of our presentation.

Re: Spectrum01563-24R1 (Integrative study of pulmonary microbiome and clinical diagnosis in Pulmonary Tuberculosis Patients)

Dear Dr. Qiwen Yang:

Your manuscript has been accepted, and I am forwarding it to the ASM production staff for publication. Your paper will first be checked to make sure all elements meet the technical requirements. ASM staff will contact you if anything needs to be revised before copyediting and production can begin. Otherwise, you will be notified when your proofs are ready to be viewed.

Sincerely,
Fei Chen
Editor
Microbiology Spectrum

Reviewer #1 (Comments for the Author):

1. Abbreviations introduced in the abstract lack full definitions, please add.
2. line 226, "multiple next-generation sequencing (mNGS)", multiple or metagenomic?

1. Abbreviations introduced in the abstract lack full definitions, please add.
2. line 226, "multiple next-generation sequencing (mNGS)", multiple or metagenomic?